# CryoEM structure of the outer membrane secretin channel pIV from the f1 filamentous bacteriophage

Rebecca Conners [1,2], Mathew McLaren[1,2], Urszula Łapińska[1,2], Kelly Sanders[1,2], M. Rhia L. Stone[3], Mark A. T. Blaskovich [3], Stefano Pagliara[1,2], Bertram Daum [1,2], Jasna Rakonjac [4] & Vicki A. M. Gold [1,2✉]

The Ff family of filamentous bacteriophages infect gram-negative bacteria, but do not cause lysis of their host cell. Instead, new virions are extruded via the phage-encoded pIV protein, which has homology with bacterial secretins. Here, we determine the structure of pIV from the f1 filamentous bacteriophage at 2.7 Å resolution by cryo-electron microscopy, the first near-atomic structure of a phage secretin. Fifteen f1 pIV subunits assemble to form a gated channel in the bacterial outer membrane, with associated soluble domains projecting into the periplasm. We model channel opening and propose a mechanism for phage egress. By single-cell microfluidics experiments, we demonstrate the potential for secretins such as pIV to be used as adjuvants to increase the uptake and efficacy of antibiotics in bacteria. Finally, we compare the f1 pIV structure to its homologues to reveal similarities and differences between phage and bacterial secretins.

[1] Living Systems Institute, University of Exeter, Exeter, UK. [2] College of Life and Environmental Sciences, Geoffrey Pope, University of Exeter, Exeter, UK. [3] Centre for Superbug Solutions, Institute for Molecular Bioscience, The University of Queensland, Brisbane, Queensland, Australia. [4] School of Fundamental Sciences, Massey University, Palmerston North, New Zealand. ✉email: v.a.m.gold@exeter.ac.uk

The Ff family of filamentous bacteriophages includes f1, fd and M13, and is one of the simplest biological systems known[1]. Filamentous phages such as Ff have many uses in biotechnology[2]; examples include as a cloning vector, and in phage display for screening protein-ligand and protein-protein interactions. Ff is also increasingly employed in nanotechnology, being engineered to adopt a variety of different forms such as nano-wires, nanorings and branched structures[3,4]. With the recent, and much publicised, global antimicrobial resistance crisis[5,6], there has been a renewed campaign in exploring the use of phages and their proteins to target pathogenic bacteria[7,8]. Understanding the structural and functional details of phage lifecycles is thus of broad interest for both fundamental and applied biological science research.

Phages f1, fd and M13 share 98.5% sequence identity and infect the gram-negative bacterium *Escherichia coli*. The primary binding receptor is the F-pilus - a long filamentous appendage assembled on the bacterial cell surface. Phage virions are 1 μm long and 6–7 nm wide, with a circular single-stranded DNA genome encoding for 11 phage proteins. The phage genome is encapsulated within several thousand copies of the major phage coat protein pVIII and is capped by protein complexes consisting of pIII/pVI and pVII/pIX at either end[9]. Phages bind to the F-pilus via the pIII/pVI cap, which is followed by pilus retraction and binding of phage to the TolQRA complex in the host cell cytoplasmic membrane[10]. The phage genome is released into the bacterial cytoplasm, DNA is replicated and phage proteins are expressed[11]. New virions are assembled and released from the cell via a complex of three phage-encoded proteins: pI, pXI and pIV, which form a trans-membrane complex[12]. pI (an ATPase) and pXI span the cytoplasmic membrane with pIV located in the outer membrane[13].

pIV is a secretin family protein, with sequence similarity to bacterial secretins at the core of the Type IV pilus assembly machinery and the Type II and Type III secretion systems[14]. A number of high-resolution structures of bacterial secretins have been determined which have allowed detailed comparisons to be made[15,16]. All secretins are multimeric channel proteins, ranging in molecular weight from 670 kDa to >1 MDa. Reports of stoichiometry range between twelve and fifteen subunits per complex, divided into a number of different domains. The C-terminal domain integrates into the bacterial outer membrane and forms a double-walled ß-barrel with a central gate forming a stricture at the centre[17]. Varying numbers of soluble N-domains connect to the ß-barrel and project into the periplasm. The N-domains are often poorly ordered in cryo-electron microscopy (cryoEM) structures, presumably due to their flexibility. Type II secretins can also differ at the externally-exposed side of the ß-barrel, with *Vibrio*-type complexes possessing a cap gate and *Klebsiella*-type lacking this feature[15]. A cryoEM structure of the f1 pIV (f1pIV) protein determined at 22 Å resolution[18] showed a barrel-like complex with a large pore running through the centre containing a pore-gate.

We present here the cryoEM structure of the closed state of f1pIV to 2.7 Å resolution, the first near-atomic resolution structure of a phage secretin protein. Based on the conserved nature of secretins, we computationally model the open state f1pIV structure and its interaction with phage. By correlating our structures to phenotypic data on f1 and f1pIV, we propose a mechanism for phage egress. We also employ a mutant of f1pIV that produces a leaky *E. coli* phenotype and demonstrate by single-cell microfluidics that the macrolide antibiotic roxithromycin is taken up efficiently by the f1pIV-expressing bacteria. The f1pIV atomic structure is used to interpret our findings, which have important translational implications for the use of secretins as therapeutic adjuvants to increase antibiotic delivery to bacteria. Finally, we reveal the common structural motifs shared by f1pIV and bacterial secretins, and differences in their electrostatic surface charges and modular architecture.

## Results

**Structure of the f1pIV secretin**. The 670 kDa f1pIV secretin channel was expressed recombinantly in *E. coli* and purified in the presence of CHAPS detergent by $Ni^{2+}$ affinity and size exclusion chromatography. Pure protein was identified by SDS-PAGE, Western blot and mass spectrometry analysis (Supplementary Fig. 1). CryoEM data were collected and processed in Relion 3.1[19] (Supplementary Table 1). Averaging the protein particles together in 2D showed that f1pIV particles are composed of fifteen identical subunits (Fig. 1a); no variation in symmetry was observed. 3D reconstruction produced a 2.7 Å map with C15 symmetry applied (Fig. 1b and Supplementary Fig. 2).

The N-terminal part of f1pIV is comprised of two periplasmic N domains: the far N-terminal N0 domain (residues 1–88) and the N3 domain (108–172), which are joined together with a 19-residue linker (89–107) (Fig. 2a). The N3 domain links to the C-terminal secretin domain (175–405) via a two amino acid linker (173–174) (Fig. 2a). The quality of our map allowed the N3 and secretin domains to be unambiguously modelled into the density (Figs. 1c and 2b, c), with clear side chain densities visible in most areas (Fig. 2d). The fifteen subunits pack against each other to form the secretin channel, with each subunit contributing a four-stranded ß-sheet to the multimer, culminating in a large ß-barrel composed of 60 ß-strands (Fig. 2b, c). The ß-sheets lie at an angle of 41° relative to the central axis of the pore. Within the ß-barrel is a smaller inner barrel comprised of four ß-strands from each f1pIV subunit (Fig. 2c, e–g). The inner ß-barrel is angled parallel to the pore axis and provides a scaffold for two extended loops that reach into the centre of the pore, known as Gate 1 and Gate 2[20] (Fig. 2c, g). The interface between the outer and inner ß-barrels is composed mostly of hydrophobic residues, and strengthened further with a salt bridge between Glu 185 and Arg 337 (both residues are highly conserved amongst homologues) (Fig. 2e, Supplementary Figs. 3 and 4). Extensive hydrogen bonds are formed in the inner and outer ß-barrels; within each individual subunit, and also between each subunit and its immediate neighbours (Fig. 2f). In the outer ß-barrel, subunit 1 forms 28 intra-subunit mainchain hydrogen bonds, and 20 inter-subunit mainchain hydrogen bonds (10 with each neighbouring subunit). In the inner ß-barrel and gate, subunit 1 forms 34 intra-subunit mainchain hydrogen bonds and 16 inter-subunit ones (8 with each neighbouring subunit). The interface between neighbouring N3 domains is mostly hydrophobic, and no hydrogen bond interactions are observed between them in our structure. For reasons of flexibility (Supplementary Fig. 2), the extracellular part of the ß-barrel was missing interpretable density (36 amino acid residues, positions 199–234) as well as the very centre of the pore in the Gate 1 loop (15 residues, positions 275–289).

The f1pIV secretin measures 96 Å from the extracellular face of the ß-barrel to the N-terminus of the N3 domain, and 112 Å across at its widest point (within the ß-barrel). The diameter across the inner ß-barrel is 90 Å. The pore width is constricted further in two areas; at the central gate (40 Å) and at the N3 domains (74 Å). There are however disordered loops in both the N3 domain and Gate 1, both pointing into the centre of the pore, so these distances could be smaller.

The globular N3 domain consists of 65 residues which fold to form a three-stranded ß-sheet packed against two α-helices, joined by the short two amino acid linker to the ß-barrel (Fig. 2a, b, c, g). The far N-terminal N0 domain and its associated 19

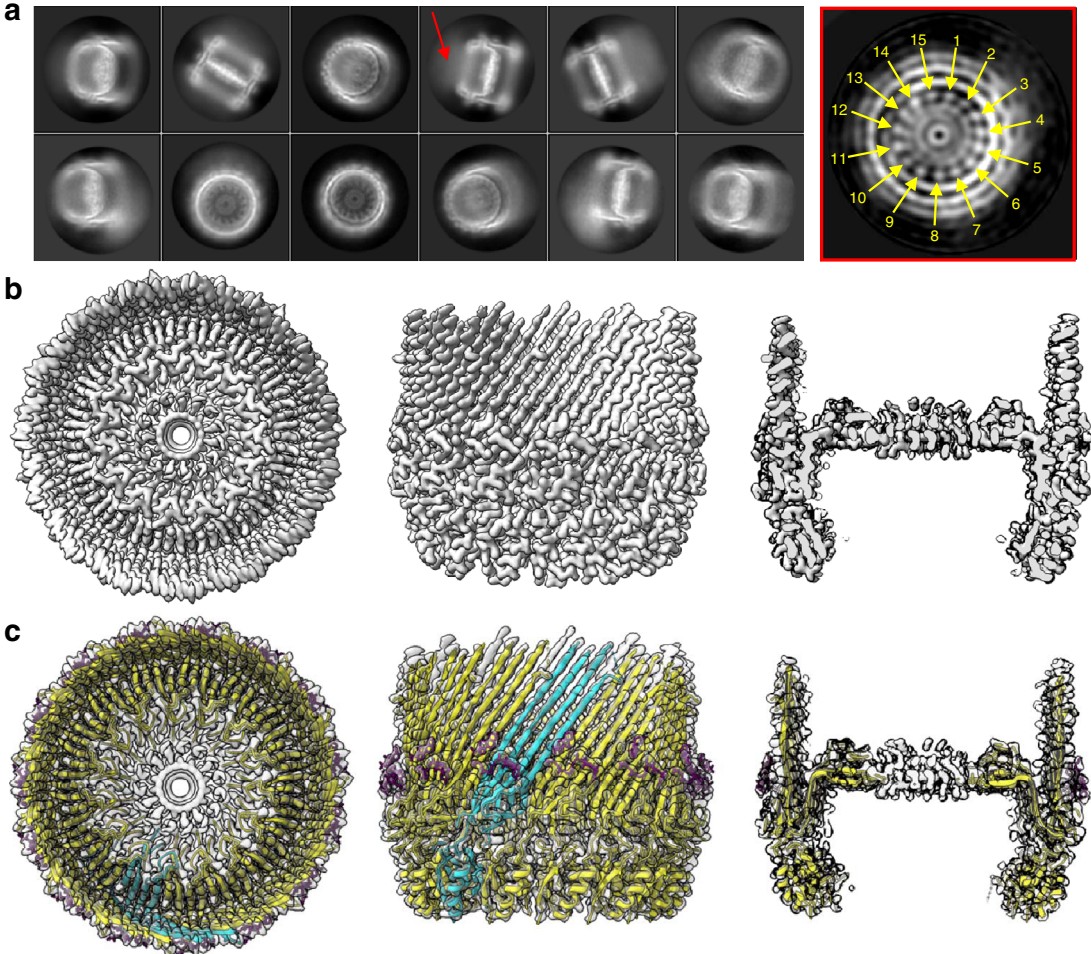

**Fig. 1 f1pIV cryoEM data and atomic model. a** Left, 2D class averages from Relion. A red arrow highlights a hazy area, likely resulting from protein flexibility. Right, a cross-section through the initial 3D model from Relion (generated without symmetry applied), highlighting the observed C15 symmetry (yellow arrows), displayed using IMOD[72]. **b** Final 3D reconstruction shown from left to right as top view, side view and side view sliced through the centre. **c** f1pIV protein structure modelled into the experimental density map shown as a top view, side view and side view sliced through the centre. A single f1pIV subunit is shown as a cyan cartoon with the remaining 14 subunits in yellow; the EM map is shown in light grey. A band of 3-[(3-cholamidopropyl) dimethylammonio]-1-propanesulphonate (CHAPS) detergent molecules is shown as purple sticks.

residue linker region (residues 1–107, Fig. 2a) could not be observed in our structure. In the 2D classes, a hazy area could be seen in the expected region, suggesting that N0 could be present but extremely flexible (Fig. 1a, red arrow). Tryptic digest mass spectrometry was used to confirm that all of the regions for which we either did not see density, or could not interpret the density, were physically present (Fig. 2a).

To provide insight into the structure of the N0 domain, we employed four complementary methods. We performed a Blast search of f1pIV N0 plus the associated linker region (residues 1–107) against all structures in the Protein Data Bank. This resulted in a single hit for the PulD secretin from *Klebsiella pneumoniae* (6HCG)[21] with 33% sequence identity over 63 residues. We used this structure to produce a homology model of the f1pIV N0 domain in Swiss-Model[22]. The same f1pIV N0-linker domain sequence was also submitted to the I-TASSER[23], Robetta[24] and AlphaFold 2[25] structural prediction servers. The resulting models from all methods were compared and seen to be in agreement for the folded domain (residues 2–71; Supplementary Fig. 5a). The N0 domain is thus most likely formed by two α-helices, flanked by a two-stranded β-sheet on one side and a three-stranded β-sheet on the other. We used the structure from I-TASSER to generate a composite model with our f1pIV

structure (Fig. 2g). Three key residues in the N0 domain have been identified as important for mediating the interactions with the periplasmic domains of pI/pXI: Met 5[26] and Glu 4 combined with Ile 69[27]. Mapping these residues to the N0 domain shows that they are predicted to lie at the face of N0 which would be directly opposed to pI/pXI in the cytoplasmic membrane (Supplementary Fig. 5b).

The four N0 models diverge at residues 72–107, which is in agreement with this region being a flexible linker (Fig. 2a, Supplementary Fig. 5a). The linker region has been modelled in a fully extended conformation and the position of the N0 domain obtained by overlaying the f1pIV structure with that of the *Klebsiella* PulD secretin (6HCG[21], Supplementary Fig. 5c); the distance between N0 and N3 could be closer if the linker takes on a more compact structure. The entire length of f1pIV (from the extracellular face of the β-barrel to the N-terminal periplasmic domain of N0) in our composite model is 220 Å. This agrees well with the distance needed for f1pIV to cross the *E. coli* periplasm (with a width of 210 Å)[28] and reach the pI/pXI proteins in the cytoplasmic membrane.

**Membrane integration of f1pIV.** An additional band of density was observed in our EM map which was unaccounted for, lying in

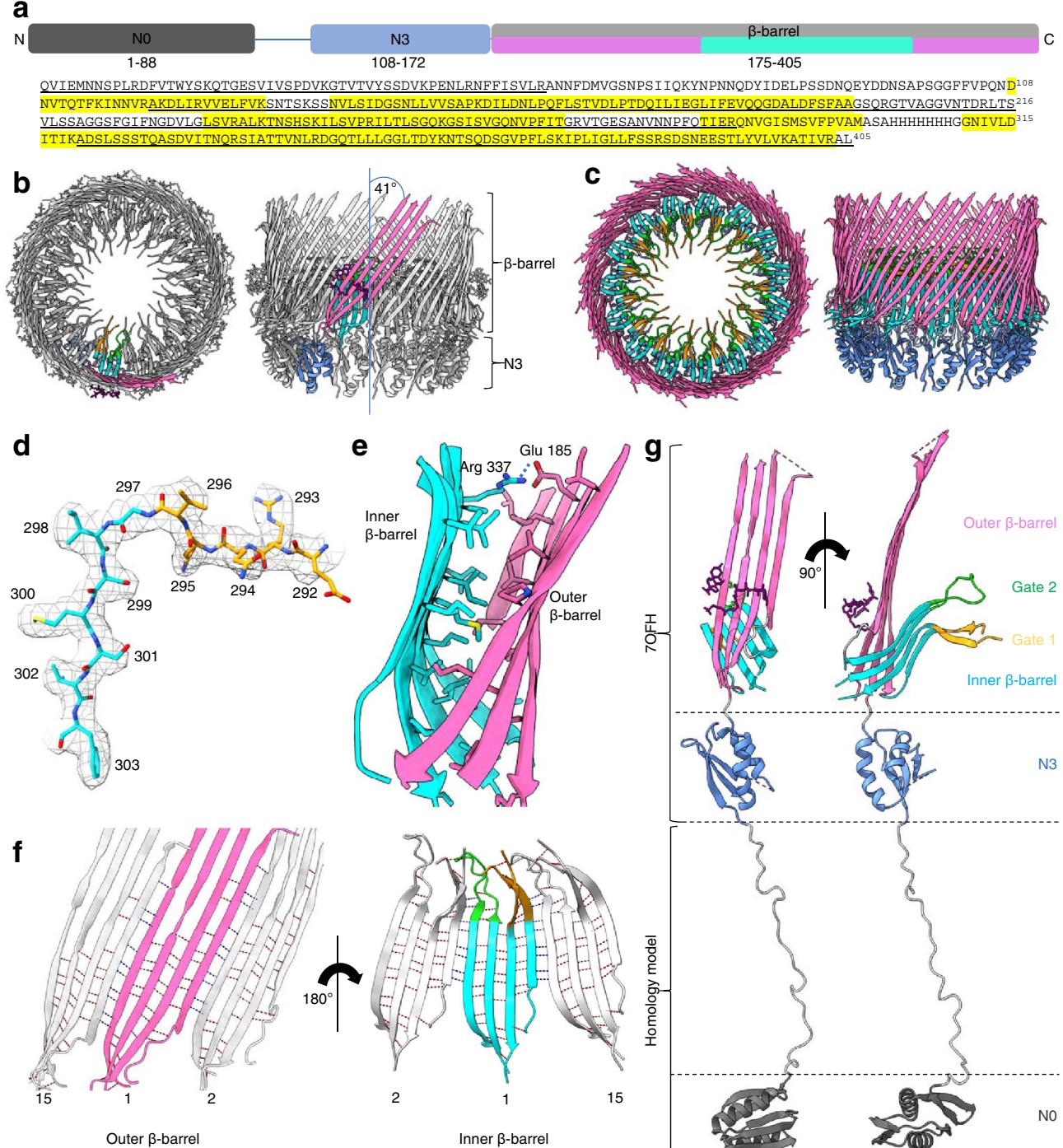

**Fig. 2 Structural details of f1pIV. a** Domain structure and amino acid sequence of f1pIV, showing the N0 domain (dark grey), the N3 domain (blue), and the β-barrel secretin domain (light grey). The regions forming the outer β-barrel are shown in pink, and that forming the inner β-barrel in cyan. Residues highlighted in yellow in the amino acid sequence were observed in the cryoEM map. Underlined residues were confirmed to be present in the cryoEM sample by mass spectrometry analysis. **b** Cartoon representation of the f1pIV multimer (top and side views) with a single molecule coloured as follows: N3 domain in blue, outer β-barrel in pink, inner β-barrel in cyan, Gate 1 in orange, Gate 2 in green, and CHAPS molecules as purple sticks. **c** Cartoon representation of the f1pIV multimer with all chains coloured as in **b**. CHAPS molecules have been removed for clarity. **d** Residues 292–303 (incorporating part of the Gate 1 loop and the inner β-barrel) showing representative density. **e** The interface between the inner and outer β-barrels is lined mostly with hydrophobic residues which are shown as sticks, and coloured by atom (N in blue, O in red, S in yellow). A salt bridge is formed at the edge of this interface between conserved residues Glu 185 and Arg 337 (dashed blue line). **f** Inter- (blue dashed lines) and intra- (red dashed lines) subunit hydrogen bonds are formed connecting the inner and outer β-barrels. One subunit (position 1) is coloured as in **b**, and the neighbouring subunits (in positions 2 and 15) are shown in grey. Only mainchain hydrogen bonds are shown; images have been clipped for clarity. **g** Composite model of a single f1pIV subunit shown as front and side views. The structure built into the EM map is coloured as in **b**, with a homology model of the N0 domain in dark grey.

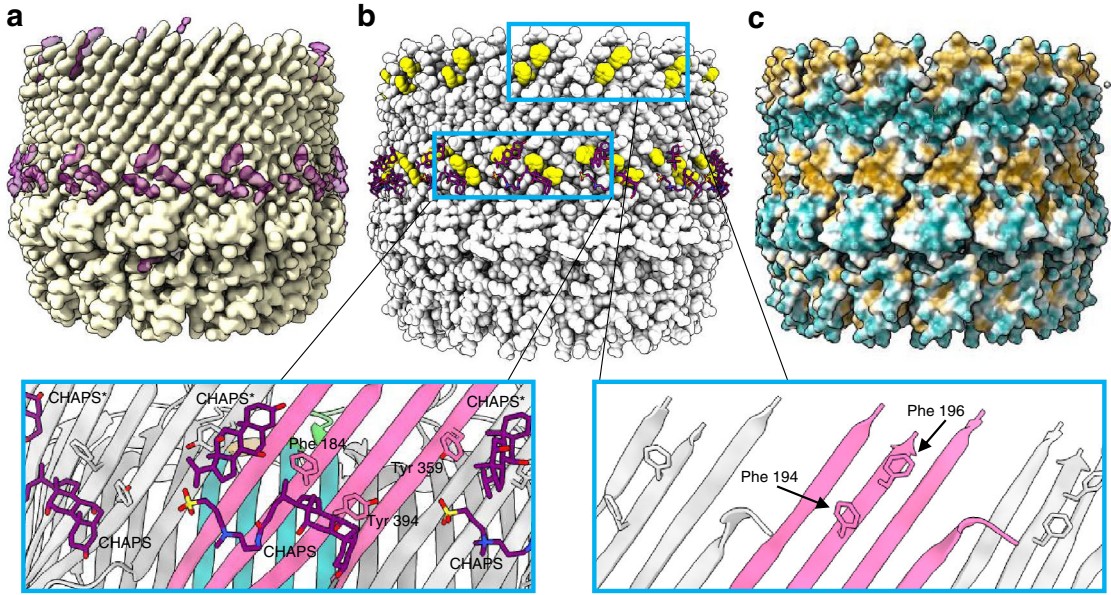

**Fig. 3 Positioning of the f1pIV multimer in the *E. coli* outer membrane. a** The f1pIV cryoEM map in side view (yellow) shows density that was unaccounted for after the atomic model had been built (purple). **b** The f1pIV atomic model is shown as space-filled atoms in light grey with the CHAPS shown as purple sticks. There are two bands of aromatic residues shown in yellow (aromatic girdles). The enlarged panels (blue boxes) show the two girdles in greater detail with the aromatic residues labelled. One full molecule of CHAPS, and one consisting only of the aromatic rings (CHAPS*) were modelled in to the ring of density. CHAPS is shown as purple sticks, and the f1pIV subunits are coloured as in Fig. 2b. **c** f1pIV atomic model shown as surface representation is coloured by hydrophobicity, ranging from the most hydrophilic areas (dark cyan), through white, to the most hydrophobic areas (orange).

approximately the centre of the surface of the ß-barrel (Fig. 3a). This formed a ring around the outside of the secretin domain, reminiscent of detergent belts typically seen in cryoEM structures of membrane proteins[29]. We modelled the detergent CHAPS, used during purification, into this density. Two molecules of CHAPS were observed to bind to each f1pIV subunit, forming a circle of detergent around the barrel (Fig. 3b). The detergent molecules were bound to a ring of aromatic residues (Phe 184, Tyr 359, and Tyr 394), forming an aromatic girdle. An additional aromatic girdle was observed nearer the extracellular region of the ß-barrel formed by Phe 194 and Phe 196. Such clusters of aromatic residues are often observed at the lipid–water interface in membrane-spanning ß-barrels of gram-negative bacteria, and allow for protein positioning in the membrane[30]. Colouring the surface of f1pIV by hydrophobicity confirms that the two aromatic girdles lie within the two main regions of high hydrophobicity on the protein surface (Fig. 3c).

**The Gate region**. The inner ß-barrel, Gate 1 and Gate 2 loops form the full gate region that projects into the pore. The Gate 1 loops extend to the centre, presumably sealing the channel (Fig. 2g). Sequence alignments show high sequence conservation for Gate 1 amongst secretin homologues, while Gate 2 is less well conserved (Supplementary Fig. 3). In both cases, the areas of highest sequence conservation are found where the gates join the inner ß-barrel, with the more variable regions lying in the loop regions which face the centre of the pore. There are two highly conserved glycine residues (Gly 267 and Gly 297 in f1pIV) found at the boundary of Gate 1, which are believed to form a hinge for gate opening[31] (Supplementary Fig. 3, Fig. 4a). An extensive hydrogen bonding network was observed within the gate region (Fig. 4a). Hydrogen bonds are formed within each gate (6 mainchain intra-gate bonds in Gate 1, and 3 mainchain intra-gate bonds in Gate 2), tethering each loop together tightly. As seen in all other secretin structures to date, the Gate 1 loop has an

unusual twisted conformation. Inter-gate hydrogen bonds are formed between neighbouring subunits, with Gate 1 from one chain forming four mainchain hydrogen bonds (including the hinge residues) with Gate 2 from the neighbouring subunit (Fig. 4a). Specific residues add further stability to the gate region. For example, a highly conserved Arg 293 in Gate 1 forms hydrogen bonds with the side chain of Asn 269 within the same gate (conserved in phage secretins), and with the backbone carbonyl oxygen of an unconserved Val 332 from Gate 2 of the neighbouring subunit (Fig. 4b, Supplementary Fig. 3). As Asn 269 is conserved mostly amongst phage but not bacterial secretins, we also analysed the role of this residue in context of the pIV structure. Asn 269 forms 4 hydrogen bonds – one each with Arg 293 and Asn 295 within the Gate 1 loop, and two with Thr 334 from the Gate 2 loop of the neighbouring subunit, demonstrating the role of this residue in promoting a high level of stability.

**Opening the f1pIV channel**. The f1pIV pore must undergo significant structural rearrangements to allow phage to egress. To provide insight into the mechanism of channel opening, we compared our f1pIV structure to existing closed and open gate structures of a homologous protein: the Type III secretin InvG from *Salmonella typhimurium* (6PEE and 6Q15)[32]. Structural alignment of the two InvG structures highlights the main differences between the open and closed states (Supplementary Fig. 6). The core of the secretin domain and the conformation of the loops remain similar; differences are seen in the position of the gate loops relative to the barrel, the ß-lip and an extended loop in the N3 domain. Both gate loops move to pack against the outer ß-barrel in the open form. The Gate 1 loop moves with a hinge-like motion, while the Gate 2 loop twists. The ß-lip at the extracellular edge of the ß-barrel also moves away from the centre of the pore in the open structure, and there is movement of an extended loop (not present in f1pIV) in the N3 domain. Our closed f1pIV structure aligns well to the InvG closed state, with a

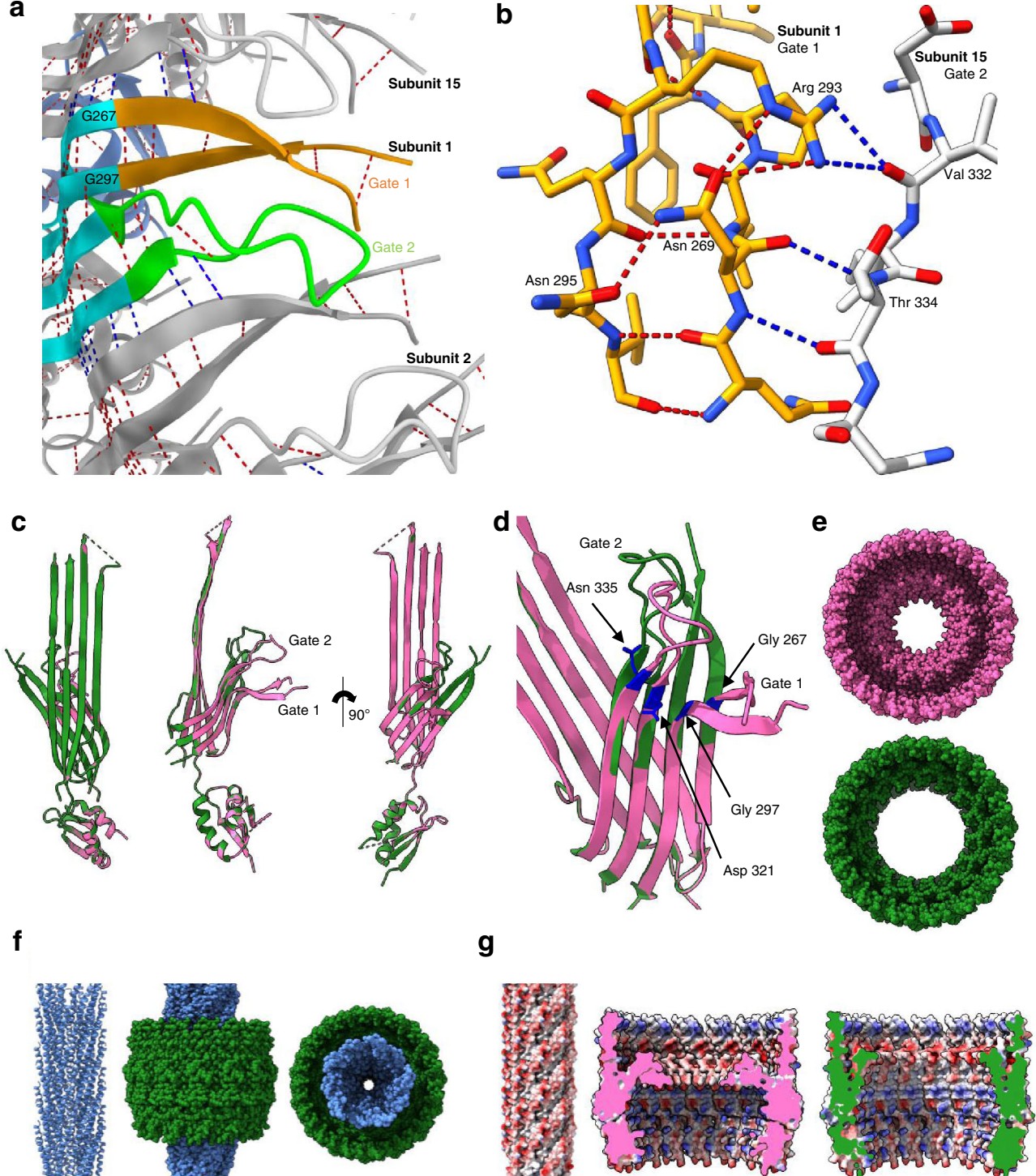

**Fig. 4 The Gate region, channel opening, and interaction with phage. a** Extensive inter- (blue dashed lines) and intra- (red dashed lines) gate hydrogen bonds are formed between neighbouring subunits and within Gate 1 and Gate 2, colouring as in Fig. 2b. Only main chain hydrogen bonds are shown. **b** Close up to show the hydrogen bonding interactions of Arg 293 and Asn 269 in Gate 1. Arg 293 forms intra-gate hydrogen bonds with Asn 269, and inter-gate hydrogen bonds with Val 332 in Gate 2 of the neighbouring subunit. Asn 269 forms intra-gate hydrogen bonds with Arg 293 and Asn 295 in Gate 1, and two inter-gate bonds with Thr 334 from the Gate 2 loop of the neighbouring subunit. Gate 1 from subunit 1 is shown in orange and Gate 2 from subunit 15 in grey, and coloured by atom (N in blue, O in red). **c** Structural superimposition of a single f1pIV subunit closed state structure (pink) with f1pIV modelled in the open gate position (green), shown in front, side and back views. **d** Close up showing the Gate 1 (Gly 267, Gly 297) and Gate 2 (Asp 321, Asn 335) hinge residues in blue. **e** The closed (pink) and open (green) multimers are shown as space-fill representation in top view. **f** Left, structure of an fd virion (2C0W) in cartoon form. Centre and right, manual docking of the fd virion (blue space-filled representation) into the open f1pIV secretin pore (green surface representation) in side and top views respectively. **g** Left, electrostatic surface potential of the fd virion coloured red for negative and blue for positive. Centre and right, electrostatic surface potential of the closed (pink) and open (green) states of the f1pIV pore, coloured as per the fd virion.

Root Mean Square Deviation (RMSD) of 1.051 Å (Supplementary Fig. 6). The f1pIV closed gate loops were thus modelled into the open form based on their position in InvG (Fig. 4c; Supplementary Fig. 6).

This model reveals how the f1pIV Gate 1 would likely pivot upwards and outwards on the highly conserved hinge residues Gly 267 and Gly 297, located at the top of the inner ß-barrel (Fig. 4a, c, d, Supplementary Fig. 3). The twisting motion in Gate 2 could occur around the conserved Asp 321 and Asn 335 (Fig. 4d, Supplementary Fig. 3). These movements would move both gate loops closer to the outer ß-barrel and thus result in an opening of the pore from the ~40 Å measured in our closed structure to 65 Å (Fig. 4e). To confirm that these dimensions are sufficient for egressing phage, we docked the fibre diffraction structure of an fd phage (2C0W)[33] into the open state of f1pIV (Fig. 4f). Colouring the phage by electrostatic potential shows a clear negative charge on the outer surface (Fig. 4g). In contrast, the electrostatic surface potential within the f1pIV pore is more varied. In closed and open states, there is a band of positive charge in the area closest to the extracellular side, followed by a band of negative charge near the gate region. In the open state, the aperture revealed by gate opening is mostly positively charged (Fig. 4g).

**Mapping mutations to f1pIV that cause a leaky E. coli phenotype.** Wild type E. coli cells (strain K12) cannot transport the bile salt deoxycholate across their outer membranes, and those which are deficient in the LamB maltoporin cannot transport large maltooligosaccharide sugars. Previous studies[20,34,35] have revealed a number of mutations in the f1pIV protein that render the pore leaky to these substrates. Atomic-level detail of the f1pIV structure can now be used to understand how these mutations cause their effect. We mapped leaky mutations discovered by random mutagenesis (Supplementary Table 2)[20] on to the f1pIV structure (Fig. 5a, b). The overwhelming majority (90%) were found to lie in the gate region (the Gate 1 and Gate 2 loops or inner ß-barrel). Three mutations (A121V, D123Y, G147V) conferring leakiness to deoxycholate and maltopentaose were identified in the N3 domain, all at the interface closest to the inner ß-barrel; one mutation (I183V) was observed in the outer ß-barrel (Fig. 5a). The authors also tested susceptibility of E. coli expressing different f1pIV mutants to the antibiotics vancomycin and bacitracin (which are too large to cross the bacterial outer membrane). By mapping these to our f1pIV structure, we can visualise directly that all mutants which conferred antibiotic susceptibility were located in the gate region (4 with sensitivity to vancomycin alone, 12 with sensitivity to both vancomycin and bacitracin) (Fig. 5b).

We used the map of f1pIV mutants to understand more about the delicate balance of interactions in the gates (Supplementary Table 2). The conserved Gate 1 hinges (Gly 267 and Gly 297) pack closely together within the inner ß-barrel (Fig. 4d). Changing Gly 267 or Gly 297 to any other sidechain in the model causes steric clashes. Hinge mutations would thus disrupt the packing in the region, interfere with hinge motion and prevent proper gate closure. Asn 269, Arg 293, and Asn 295 in Gate 1 take part in a tight hydrogen bonding network in the closed structure (Fig. 4b) that differs to the other secretins analysed (e.g. in Fig. 6). Mutations in any one of these residues results in leaky f1pIV (Fig. 4b; Supplementary Table 2, Supplementary Fig. 7). We can now rationalise how these mutations lead to a more open gate structure—by disrupting the hydrogen bonding network within the Gate 1 loop and between two neighbouring gates.

**Antibiotic uptake through an f1pIV leaky mutant.** The antibiotic vancomycin is hydrophilic and bacitracin is amphipathic;

both are sufficiently small to pass through the open f1pIV pore (Supplementary Fig. 8). E. coli expressing the previously characterised f1pIV mutant S324G are sensitive to both antibiotics[20,36]. Ser 324 is located in Gate 2, where mutation to a glycine will destroy the hydrogen bond made between the Ser 324 sidechain and the backbone nitrogen of Ser 326 (Fig. 5c). Mutation to a glycine could also change the position of the peptide backbone, potentially altering the hydrogen bonds that are currently made between the Ser 324 carbonyl oxygen and the sidechain of Gln 328 from the neighbouring subunit, and those made between the backbone nitrogen of Ser 324 and the carbonyl oxygen of Ile 333 (Fig. 5c). The combined effect of these mutations likely results in increased mobilisation of the gate and improved antibiotic uptake.

Here we aimed to understand whether the uptake and efficacy of hydrophobic antibiotics could be improved by expressing leaky f1pIV in E. coli. Hydrophobic macrolides are often used as first-line antibiotics, however, they are used typically to treat infections caused by gram-positive but not most gram-negative bacteria due to slow uptake across the outer membrane[37]. The macrolide antibiotic roxithromycin is widely clinically employed[38], and our modelled open gate structure reveals that it is sufficiently small to pass through the open f1pIV channel unhindered (Supplementary Fig. 8). Using single-cell analysis with the microfluidic mother machine and time-lapse microscopy[39,40], bacteria were injected into channels and the accumulation of fluorescent antibiotics measured[41]. Specifically, we employed a recently introduced fluorescent derivative of roxithromycin (roxithromycin linked to nitrobenzoxadiazole-roxithromycin-NBD), which largely maintains the antibiotic potency of the parental drug[37]. Accumulation of roxithromycin-NBD was significantly higher in E. coli cells expressing f1pIV[S324G] compared to wild type f1pIV (Fig. 5d). We also measured the minimum inhibitory concentration (MIC) for roxithromycin and determined a one-notch shift against the mutant compared to the wild-type (64 and 96 μg/ml, respectively). This indicates that the higher accumulation of roxithromycin-NBD corresponds to a higher efficacy of the antibiotic in cells expressing leaky f1pIV[S324G].

**Structural comparison of f1pIV with other secretins.** As our structure is the first of a phage secretin at near-atomic level, we compared f1pIV to different classes of bacterial secretins[31,32,42] (Fig. 6a–c, Supplementary Fig. 9). All secretins share the same overall architecture of the double-walled ß-barrel and the first and last periplasmic domains (N0 and N3), but differ in the number of additional N domains (reviewed by Filloux and Voulhoux[15]). The Gate 1 and Gate 2 loops are all in similar conformations, pointing into the centre of the pore from the inner ß-barrel. The N3 domains are broadly similar, with the one notable difference being an extended loop region in the Type III InvG secretin which extends up towards the central gate area (Fig. 6c; Supplementary Fig. 9). In general, there is increasing disorder within domains closest to the N-terminus, and many structures have no clear density for their N0 domains.

Differences between secretins are seen on the extracellular face of the ß-barrel with the Vibrio-type group of Type II secretins possessing a cap gate which forms a much narrower opening when compared to the other open-edged ß-barrels. From sequence analysis (Supplementary Fig. 4), we believe f1pIV will not have a cap structure at the extracellular side of the ß-barrel, but we could not see this region in our cryoEM map, likely for reasons of flexibility (Supplementary Fig. 2). We are missing density for 36 residues, which is the typical size of the ß-lip domain in phage secretins (Supplementary Fig. 4). In contrast, the bacterial Type II secretins typically have 51–60 residues in the ß-lip region, while

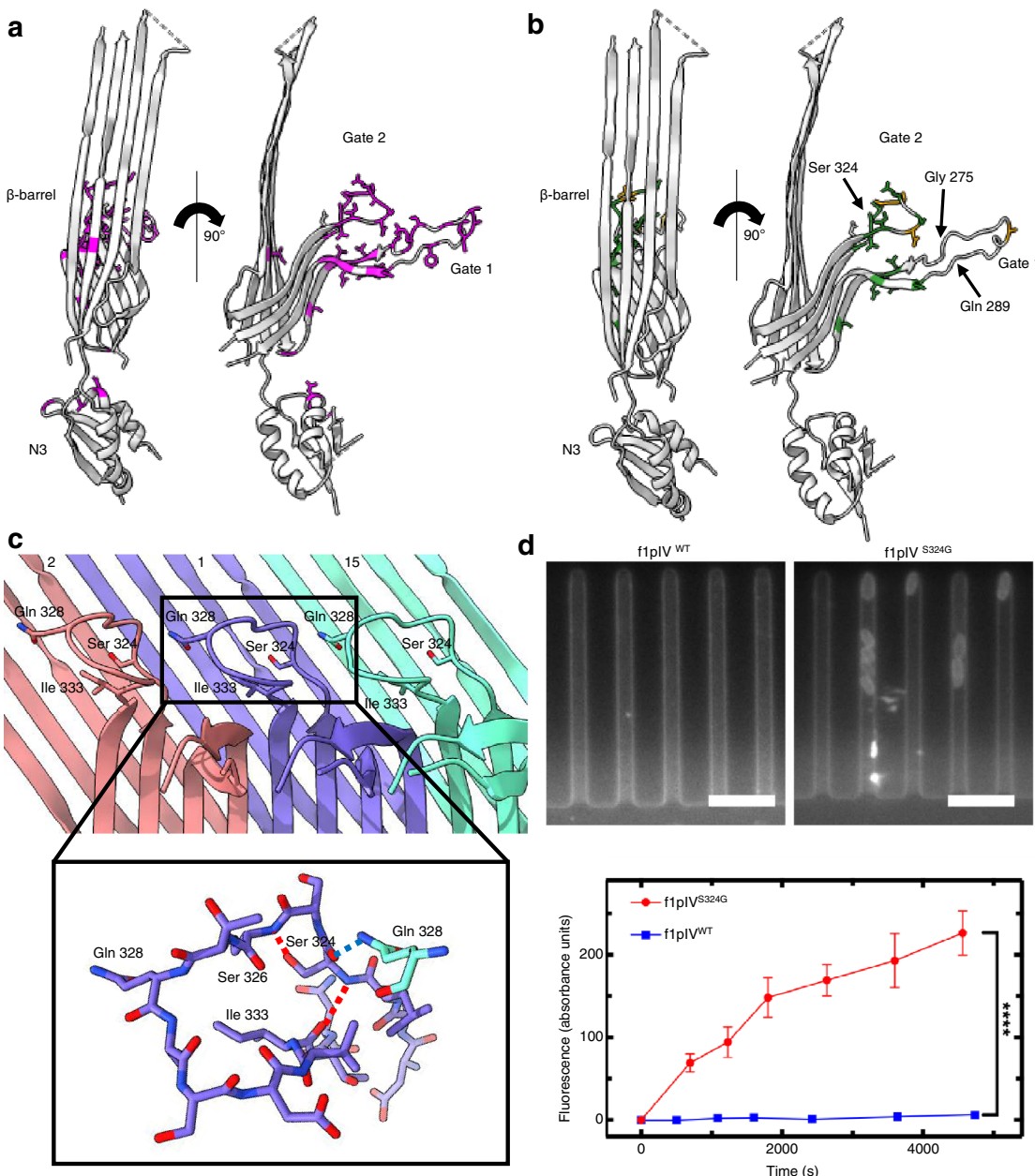

**Fig. 5 Antibiotic susceptibility in *E. coli* expressing leaky mutants of f1pIV. a** Front and side views showing all mutants[20] that were leaky to both maltopentaose and deoxycholate as magenta sticks. **b** Front and side views showing the mutations which caused sensitivity to the antibiotics vancomycin and bacitracin as green sticks. Mutations which caused sensitivity to vancomycin only are shown as orange sticks. In parts **a** and **b** the 15-residue loop from residues G275-Q289 (labelled), which is disordered in our structure, has been modelled in for representation purposes. The position of the Ser 324 mutation to Gly (S324G), analysed in part **d**, is also shown. **c** Gate 2 loop showing the key hydrogen bonds made by Ser 324. Subunit 1 is shown in purple, with neighbouring subunits 2 and 15 in red and green respectively. The close up shows that Ser 324 forms intra-gate hydrogen bonds with Ile 333 and Ser 326 (red dashed lines), and an inter-gate hydrogen bond with Gln 328 from a neighbouring subunit (blue dashed line) (N in blue, O in red). **d** Top, individual *E. coli* cells expressing f1pIV^S324G take up fluorescently labelled roxithromycin. No fluorescence change was observed for the cells expressing wild type f1pIV (f1pIV^WT). Scale bar, 5 µm. Bottom, scatter graph depicting the fluorescence change for f1pIV^WT and f1pIV^S324G samples, confirming roxithromycin uptake in the S324G mutant. Data were analysed using an unpaired t-test with Welch's correction. **** = $p \leq 0.0001$. $n = 241$ biologically independent cells for f1pIV^WT and $n = 260$ for f1pIV^S324G; error bars represent standard error of the mean.

those with a cap gate have around 80 residues here. Phage secretins are therefore expected to have a more compact ß-lip structure than their bacterial counterparts.

At the far C terminus of the protein, the Type II and Type III secretins have a helical domain (the S-domain) that packs against the outside wall of the outer ß-barrel (Fig. 6c). The S-domain binds pilotin molecules, which are essential for the assembly and/or localisation of secretins to the outer membrane. f1pIV and the

Type IV pilus secretins do not have an S-domain, with their respective C-termini being located at the periplasmic side of the outer ß-barrel (Fig. 6c). Interestingly, the Type IV pilus secretins bind to their pilotin partner via a hydrophobic girdle around the ß-barrel, in a similar place to the central hydrophobic girdle that we have observed in f1pIV (Supplementary Fig. 10).

We used the Consurf server[43] to perform a sequence alignment of f1pIV homologues, map them to the f1pIV structure and

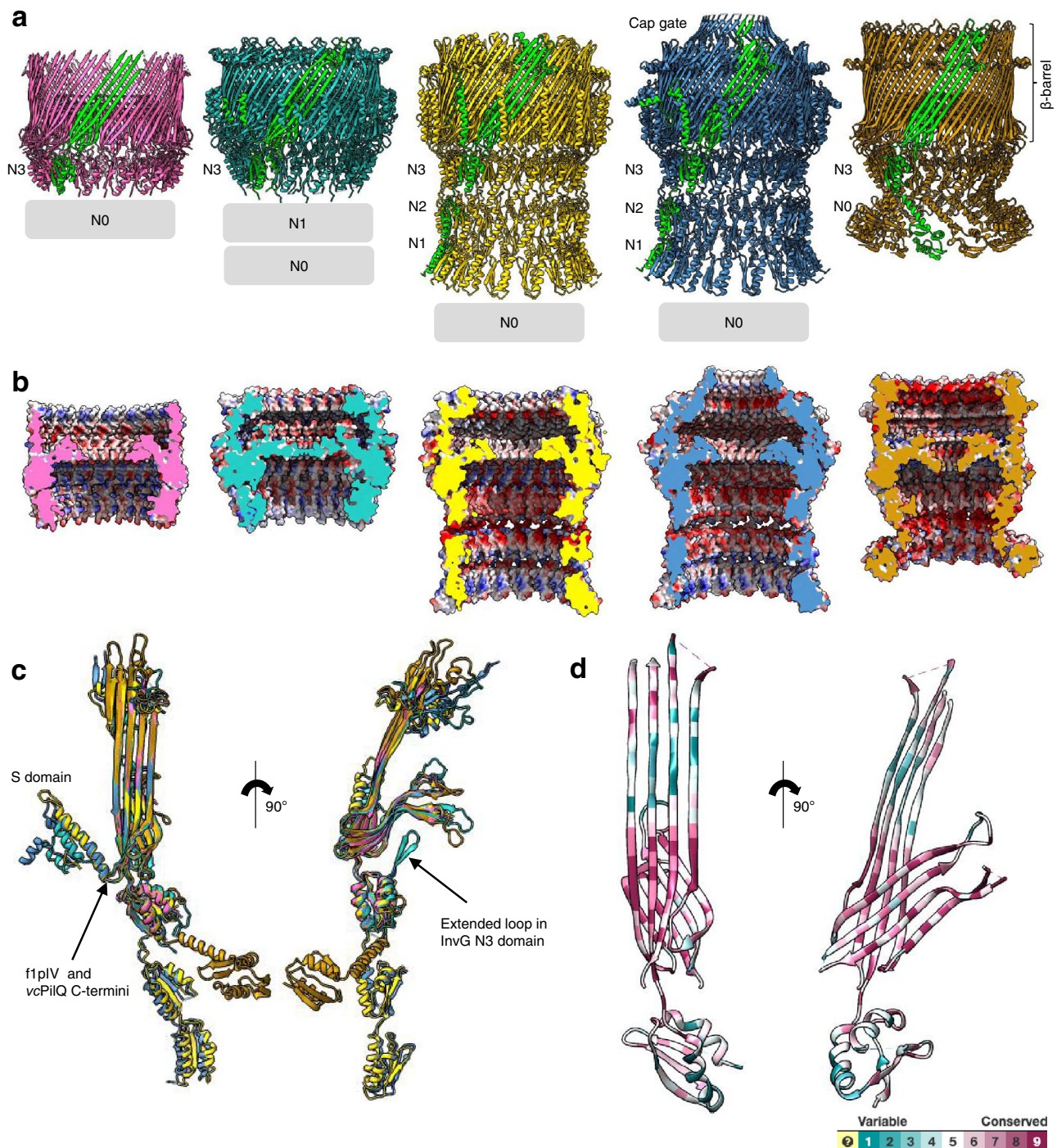

**Fig. 6 Structural comparison of f1pIV with a selection of different classes of bacterial secretins.** Secretins are aligned at the β-barrel domain. f1pIV is shown in pink (this study:7OFH), the Type III secretin InvG from *Salmonella typhimurium* in sea green (6PEE), the Type II secretin GspD from *E. coli* K12 in yellow (*Klebsiella*-type, 5WQ7), the Type II secretin GspD from *Vibrio cholerae* in blue (*Vibrio*-type, 5WQ8) and the Type IV pilus secretin PilQ from *Vibrio cholerae* in orange (6W6M). The secretins are shown in **a**) as multimers (with one subunit coloured lime green). Periplasmic N domains not observed in the maps are shown as grey boxes. **b** Electrostatic surface potential comparison of multimers cut-through to show the inner surface inside the pore. **c** Structural superimposition of the selected secretin subunits (shown individually in Supplementary Fig. 9) in front (left) and side (right) views. **d** Sequence conservation amongst the secretin family plotted on to the f1pIV structure, calculated using Consurf and based on 100 unique homologues from the UniProt database sharing 35–100% identity.

colour them by the degree of sequence conservation (Fig. 6d). The area of greatest conservation lies within the inner ß-barrel and the region of outer ß-barrel that packs against it, demonstrating the importance of this region for protein stability and/or function. The N3 domain is more variable, likely as a result of evolution to

suit the function of the individual system. We also compared the electrostatic surface potential between the different classes of secretins (Fig. 6b and Supplementary Fig. 9). The outside surfaces have a mixture of positively and negatively charged areas, with some very clear bands of hydrophobicity, as expected for outer

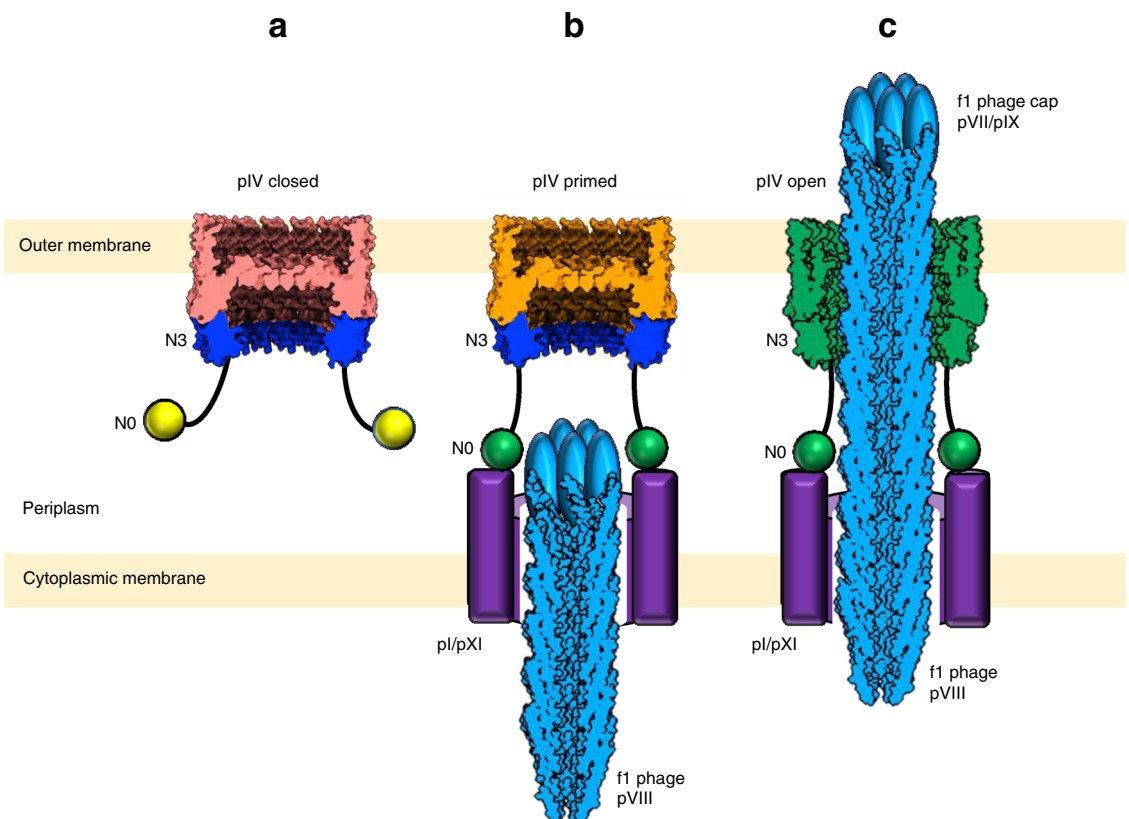

**Fig. 7 Model for filamentous phage egress. a** f1pIV with the gate in the closed state (salmon pink). The N3 (blue) and N0 (yellow) domains project into the periplasm. N0 is attached to N3 via a long linker and is flexible. **b** N0 (now green, activated) is bound to the cytoplasmic membrane complex of pI/pXI (purple) with pIV (orange) primed to open. **c** Egressing phage (blue) interacts with the N3 domain (now green, activated), triggering opening of pIV (now green, open) by rearrangement of the gate. The pVIII phage capsid was drawn using the 2COW structure, and the pVII/pIX cap added as a cartoon.

membrane proteins. The surface charge inside the secretins is quite varied; differences presumably arose to accommodate the particular shapes and electrostatic charges of substrate that each secretin transports.

## Discussion

f1pIV is the first near-atomic resolution example of a phage secretin. The protein has a similar overall architecture to bacterial secretins, but differs in its electrostatic surface potential inside the pore and in the number of N domains. f1pIV is relatively compact, with no cap gate, no C-terminal S-domain, and one N3 and one N0 domain; an arrangement which could have arisen to save valuable space in the limited capacity of the phage genome.

The phenotypes of many previously characterised mutants can be rationalised with our structure. f1pIV shows extensive hydrogen bonding within the outer and inner ß-barrel domains, as well as hydrophobic interactions and a salt bridge between the inner and outer walls. Mutation of two conserved hydrophobic residues at the interface to glycines (I312G and I314G) prevents multimer formation[20]. In addition, inclusion of a His-tag on the external loop between the inner and outer ß-barrels has a destabilising effect, but this can be rescued by introducing a hydrophobic mutation (S318I)[44], which lines the interface. The conserved hydrophobic residues between inner and outer ß-barrels are clearly important for folding and stability, and highlight the importance of the unusual double-walled architecture.

In order for the phage to pass through f1pIV, the central gate must open. Our findings, alongside f1pIV phenotypic data and the conserved architecture of the secretin family, can be used to propose a mechanism (Fig. 7). Our data show that the

periplasmic N-terminal N0 domain is extremely dynamic. This is likely due to the absence of the pI/pXI cytoplasmic membrane partners, which interact with the N0 domain of f1pIV on phage egress[12,26,27]. This flexibility may allow f1pIV to sample a wider amount of periplasmic space than a rigid structure, allowing the N0 domains to locate and bind to the cytoplasmic membrane assembly proteins. Bearing in mind that pI/pXI interaction with f1pIV is required for phage assembly[12,26], this suggests that gate opening may be triggered by interaction of the N0 domain of f1pIV with the pI/pXI cytoplasmic membrane components. Both ATP and the proton motive force are required to assemble phage[12,45], and phages with mutations in the pI protein Walker A motif are non-functional for assembly[12]. It is thus tempting to speculate that interaction of the pIV N0 domain with the periplasmic domain of pI/pXI could result in energy transfer from the pI ATPase to help drive phage egress. The egressing phage will subsequently encounter the N3 domain. Introducing mutations into the N3 domain of f1pIV causes partial gate opening[20], highlighting its role in transducing information from the periplasmic N domains into the ß-barrel. The involvement of N3 in channel opening has also been predicted for Type II and Type III secretins[31,32,46,47].

Our model of the open state of f1pIV highlights how the central gate loops will likely move to allow phage to egress via conserved hinge residues. A 10 residue in-frame deletion in Gate 2 results in leaky multimers that are non-functional for phage assembly and egress[20], suggesting that passage of phage through f1pIV is not a passive process. It is plausible that interaction of the f1 cap proteins pVII/pIX with the f1pIV gate will be required to fully open the pore, assisted by the force of the assembling f1 phage. The hydrogen bonding that we observe within Gate 1 and

Gate 2 means that the loops do not alter their conformation significantly, rather both loops are pivoted around their hinges. The hydrogen bonds between neighbouring subunits means that all gates would move synchronously.

The surface of the phage (comprised of pVIII) is negatively charged, whereas open f1pIV contains a mixture of positive and negative charge, with mostly positively charged residues in the periplasmic chamber. f1pIV can tolerate a degree of variation with respect to the pVIII coat of exported phage. For example, f1 phages with short peptides linked to the surface-exposed N-terminus are produced with similar efficiency as the wild-type phage[48]. There are however key residues that may be important in maintaining specific electrostatic interactions. For example, deletion of a predominantly negatively charged sequence (Gly3 Asp4 Asp5) from pVIII results in f1 phage that is defective in assembly[48]. In addition, the leaky mutant f1pIV$^{E292K}$ exhibits the highest sensitivity to antibiotics and a 10-fold decrease in phage assembly relative to wild-type[20]. The negatively charged sidechain of Glu 292 points into the lower cavity of the pore (Supplementary Fig. 7). Replacing this with a positively charged lysine residue would alter the electrostatic potential in this area, potentially interfering with phage assembly. It is therefore likely that a mixture of both attractive and repulsive forces are important for phage assembly and egress, similar to the findings reported for the assembly and rotation of the bacterial flagellar rod[49].

f1pIV shows most sequence similarity to GspD from *Klebsiella pneumoniae* (Supplementary Fig. 11), and thus it is plausible that the ancestral f1 phage captured a Type II secretin which subsequently evolved to meet the needs of phage assembly and egress. Indeed, a number of filamentous phages do not encode a secretin, but rather hijack one from their bacterial host[50,51]. We predict that the outermost hydrophobic girdle in f1pIV forms the membrane embedded domain, as has been proposed for other secretin structures[15,16,42,52]. The other hydrophobic girdle (located half way down the barrel) would then lie within the periplasm, in a similar position to where pilotins are seen to bind to other secretins (Supplementary Fig. 10)[53]. The fact that this hydrophobic girdle has been evolutionarily maintained in f1pIV, even in the absence of a phage pilotin, highlights its functional importance. In support of this, the C-terminus of f1pIV is in close vicinity to the girdle, and C-terminal His-tags result in non-functional f1pIV[44]. In addition, chimeras of f1pIV with C-terminal pilotin-binding S-domains from other secretins (PulD and InvG) are only targeted to the outer membrane and functional for phage assembly in the presence of their respective pilotins (PulS or InvH)[54,55]. The C-terminus and hydrophobic girdle are thus an important interaction surface, with one possibility being that the phage system is able to hijack bacterial pilotins, or pilotin-like proteins, to aid its targeting and assembly.

Uptake of hydrophilic and amphipathic antibiotics have been tested with f1pIV previously, whereas hydrophobic drugs have not. We demonstrate that the leaky mutant f1pIV$^{S324G}$ allows *E. coli* to accumulate the hydrophobic macrolide roxithromycin to a greater extent than its wild type counterpart. Ser 324 mutation to Gly likely destroys important hydrogen bonds and alters the position of the peptide backbone, resulting in a more open gate structure that is non-selective to the hydrophobicity of the molecule passing through. These findings, along with our f1pIV atomic model and the conserved nature of secretins, supports targeting the secretin family as an antimicrobial approach[56,57]. For example, in silico and in vivo screens could be used to identify molecules that can destabilise the gates, in effect sensitising any secretin-expressing pathogen to a wide range of antibiotics. Phage therapy is also receiving renewed attention, and filamentous phages have been engineered to carry lethal genes to

bacteria[58–60]. Leaky f1pIV could be cloned into the genome of phages of interest, and on protein expression channels would open in the host bacterial outer membrane, allowing antibiotics to enter. Both of these approaches would add much needed artillery to our library of therapeutic tools needed to combat the increasing rise of antimicrobial resistance.

## Methods

The system of amino acid numbering used throughout is for the mature wild-type f1pIV protein (minus the 21 signal sequence residues). Standard reagents were purchased from Sigma-Aldrich unless otherwise stated.

**Bacterial strains**. All bacterial strains used were *E. coli* K12 derivatives. TG1 electrocompetent cells were purchased from Lucigen and used for expression of f1pIV for cryoEM studies. Strain K2040 [MC4100 Δ*lamB106 degP41* (ΔPstl-Km$^R$)] described in ref. [20] was used for the microfluidics experiments.

**Expression and purification of f1pIV**. His-tagged f1pIV was recombinantly expressed in *E. coli* and purified in the presence of CHAPS detergent (Melford) based on previously published methods[36,61]. The wild type protein was engineered to have a 9 residue His tag (SAHHHHHHH) inserted at position 308 and a stabilizing S318I mutation[44]. Protein expression was induced with 1 mM IPTG overnight at 20 °C and cells were lysed by sonication. The membrane fraction was solubilized in 5% (w/v) CHAPS, 50 mM Tris pH 7.6, 500 mM NaCl, 30 mM imidazole and incubated with Ni$^{2+}$ Sepharose beads. f1pIV was eluted with a 0.03–1 M imidazole gradient, and then further purified by gel filtration chromatography on a Superose 6 Increase 10/300 GL column (GE Healthcare) in 1% (w/v) CHAPS, 25 mM Na HEPES pH 8.0, 500 mM NaCl, 0.5 mM EDTA. Purified f1pIV was separated on an Any kD mini-protean TGX gel (Biorad), electrophoretically transferred to PVDF membrane and incubated with an anti-pIV antibody at 1:2000 dilution, followed by secondary goat anti-rabbit IgG (H + L)-HRP conjugate (Biorad) at 1:3000 dilution, which was detected by Clarity$^{TM}$ Western ECL substrate (Biorad).

**CryoEM grid preparation and data collection**. The f1pIV purified sample (3 μl of ~0.7 mg/ml) was applied to graphene oxide-coated lacey carbon grids, 300 mesh (Agar Scientific) without any glow discharge, and frozen in a Mark IV Vitrobot (Thermo Fisher Scientific, 4 °C, 100% relative humidity, blot force 0, blot time 4 s). Micrographs were collected on a 300 kV Titan Krios microscope (Thermo Fisher Scientific) with a K3 direct electron detector (Gatan) at the Electron Bio-imaging Centre (eBIC) at Diamond Light Source, UK. Data were collected using EPU software (Thermo Fisher Scientific) with a pixel size of 1.072 Å (0.536 Å super-resolution) and a defocus range from −2.5 μm to −1.3 μm. Further details are shown in Supplementary Table 1. A subset of data was collected in normal mode (dataset 1), and a second subset in super-resolution mode (dataset 2).

**CryoEM image processing**. Warp[62] was used for motion correction, CTF correction and particle picking. 241,591 particles were picked from 7,037 micrographs for dataset 1 and 330,389 from 14,336 for dataset 2. Both sets of data were processed separately and combined in the latter stages of data processing. Several rounds of 2D classification and 3D refinement were implemented in Relion[19] followed by CTF refinement and polishing steps. The statistics of data collection and model reconstruction are shown in Supplementary Table 1, and data quality checks[63,64] are shown in Supplementary Fig. 2.

**Model building and refinement**. A homologous structure of the Type II secretin GspD from enteropathogenic *Escherichia coli* (5W68)[52] was manually placed in the cryoEM map using Chimera[65]. Subsequent building and model adjustments were performed using Coot[66]. Bulky residues, glycine residues, and unique sequence patterns were used to guide sequence assignment during model building. 15-fold symmetry was applied in Chimera, and the multimeric structure was refined using Refmac from the CCPEM suite[67], with the quality of the model evaluated using the validation tools in Coot and Molprobity[68]. DeepEMhancer[69] was used for denoising and postprocessing of the maps, which were then analysed to check for additional information that could be modelled from the improved maps. A Fourier Shell Correlation (FSC) curve showing the quality of the model to map fit (calculated with Phenix)[64] is shown in Supplementary Fig. 2.

**Mass spectrometry analysis**. A sample of f1pIV was run on an SDS-PAGE gel (Any kD mini-protean TGX, Biorad), stained with Coomassie, and the band of interest excised with a sterile scalpel blade. The gel band was subjected to in-gel tryptic digestion using a DigestPro automated digestion unit (Intavis Ltd.) and the resulting peptides were fractionated using an Ultimate 3000 nano-LC system in line with an Orbitrap Fusion Tribrid mass spectrometer (Thermo Scientific). In brief, peptides in 1% (vol/vol) formic acid were injected onto an Acclaim PepMap C18 nano-trap column (Thermo Scientific). After washing with 0.5% (vol/vol)

acetonitrile 0.1% (vol/vol) formic acid peptides were resolved on a 250 mm × 75 μm Acclaim PepMap C18 reverse phase analytical column (Thermo Scientific) over a 150 min organic gradient, using 7 gradient segments (1–6% solvent B over 1 min., 6–15% B over 58 min., 15–32% B over 58 min., 32–40% B over 5 min., 40–90% B over 1 min., held at 90% B for 6 min and then reduced to 1% B over 1 min.) with a flow rate of 300 nl min$^{-1}$. Solvent A was 0.1% formic acid and Solvent B was aqueous 80% acetonitrile in 0.1% formic acid. Peptides were ionized by nano-electrospray ionization at 2.2 kV using a stainless-steel emitter with an internal diameter of 30 μm (Thermo Scientific) and a capillary temperature of 250 °C.

All spectra were acquired using an Orbitrap Fusion Tribrid mass spectrometer controlled by Xcalibur 2.1 software (Thermo Scientific) and operated in data-dependent acquisition mode. FTMS1 spectra were collected at a resolution of 120,000 over a scan range (m/z) of 350–1550, with an automatic gain control (AGC) target of 400 000 and a max injection time of 100 ms. Precursors were filtered according to charge state (to include charge states 2–7), with monoisotopic peak determination set to peptide and using an intensity range from 5E3 to 1E20. Previously interrogated precursors were excluded using a dynamic window (40 s ± 10 ppm). The MS2 precursors were isolated with a quadrupole mass filter set to a width of 1.6 m/z. ITMS2 spectra were collected with an AGC target of 5000, max injection time of 50 ms, and HCD collision energy of 35%.

The raw data files were processed and quantified using Proteome Discoverer software v1.4 (Thermo Scientific) and searched against the UniProt *Escherichia coli* database (4349 sequences) plus the supplied amino acid sequence using the SEQUEST algorithm. Peptide precursor mass tolerance was set at 10 ppm, and MS/MS tolerance was set at 0.8 Da. Search criteria included carbamidomethylation of cysteine (+57.0214) as a fixed modification and oxidation of methionine (+15.9949) as a variable modification. Searches were performed with full tryptic digestion and a maximum of 1 missed cleavage was allowed. The reverse database search option was enabled and all peptide data were filtered to satisfy false discovery rate (FDR) of 5%.

**Sequence alignments and homology modelling**. The sequence of the N terminal region of the model that was not accounted for in our structure (residues 1–107) was subjected to a Blast search[70] against all sequences in the Protein Data Bank. A sequence alignment of the top hit was performed on f1pIV with Praline[71] and Swiss-Model[22] used to create a homology model of the f1pIV N0 domain. In addition, the sequence of the f1pIV N0 domain was submitted to the I-TASSER structural prediction server[23], to the Robetta structural prediction server[24], and to the Alphafold 2 structural prediction server[25]. The top predicted models for all methods were in agreement for residues 2–71 where they started to diverge. Open and closed gate structures of InvG from *Salmonella typhimurium* were super-imposed with the closed structure of f1pIV, and Gate 1 and 2 loops were modelled into an open position using Coot. 15-fold symmetry was applied to produce a model of the f1pIV open state.

The Consurf server[43] was used to perform a sequence alignment of f1pIV homologues (101 unique sequences with 35–100% sequence similarity from the UniProt database were used for this analysis), map them to the f1pIV structure and colour them by the degree of sequence conservation.

**Determination of minimum inhibitory concentration**. *E. coli* cells were freshly transformed with plasmid pPMR132$^{WT}$ or pPMR132$^{S324G}$ and single colonies of *E. coli* f1pIV$^{WT}$ or *E. coli* f1pIV$^{S324G}$ were picked and cultured at 37 °C in FB media (2.5% (w/v) tryptone, 0.75% (w/v) yeast extract, 0.6% (w/v) NaCl, 0.1% (w/v) glucose, 50 mM TrisHCl (pH 7.5)). Plasmid DNA was miniprepped and sequenced for each experiment to ensure there had been no additional mutations. Protein expression was induced overnight (in FB media plus 0.5 mg/ml bovine serum albumin) with 1 mM IPTG at 20 °C, then diluted 40-fold and grown to OD$_{600}$ = 0.5. 60 μl of roxithromycin stock (640 μg/ml in DMSO) were added to the first column of a 96-well plate. 40 μl of the induction media (FB media, 1 mM IPTG, 0.5 mg/ml BSA) was added to the first column, and 30 μl to all other wells. A total of 70 μl solution was then withdrawn from the first column and serially transferred to the next column until 70 μl solution withdrawn from the last column was discharged. The mid-log phase cultures (i.e. OD$_{600}$ = 0.5) were diluted to 10$^6$ colony forming units (c.f.u.)/ml and 30 μl was added to each well, to give a final concentration of 5 × 10$^5$ c.f.u./ml. Each plate contained two rows of 12 positive control experiments (i.e. bacteria growing in induction media without roxi-thromycin) and two rows of 12 negative control experiments (i.e. induction media only). Plates were incubated at 37 °C overnight. The minimum inhibitory con-centrations (MICs) of roxithromycin against *E. coli* f1pIV$^{WT}$ and *E. coli* f1pIV$^{S324G}$ were determined visually, with the MIC being the lowest concentration well with no visible growth (compared to the positive control experiments).

**Single-cell microfluidics**. Single-cell microfluidics experiments to measure the intracellular accumulation of roxithromycin-NBD were carried out as previously reported[37,39,41]. Briefly, overnight cultures were prepared as described above and a 50 ml aliquot was centrifuged for 5 min at 2,600 × *g* at 37 °C and resuspended at an OD$_{595}$ of 75. A 2 μl aliquot of this suspension was injected in the microfluidic mother machine device and incubated at 37 °C. The microfluidic device was completed by the integration of fluorinated ethylene propylene tubing (1/32″ × 0.008″). The inlet

tubing was connected to the inlet reservoir which was connected to a computerised pressure-based flow control system (MFCS-4C, Fluigent). The microfluidic device was mounted on an inverted microscope (IX73 Olympus, Tokyo, Japan) and the main microchamber of the mother machine was washed into the waste reservoir by flowing media at 100 μl/h for 2 h. Images were collected via a 60×, 1.2 N.A. objective (UPLSAPO60XW, Olympus) and a sCMOS camera (Zyla 4.2, Andor, Belfast, UK). After this initial 2 h growth period in growth media, the microfluidic environment was changed by flowing media containing roxithromycin-NBD at a concentration of 46 μg/ml. Upon acquiring a bright-field image the microscope was switched to fluorescent mode and FITC filter using custom built Labview software. A fluores-cence image was acquired by exposing the bacteria to the blue excitation band of a broad-spectrum LED (CoolLED pE300white, Andover, UK) at 20% of its intensity. Bright-field and fluorescence imaging was carried out every 10 min. The entire assay was carried out at 37 °C in an environmental chamber (Solent Scientific, Portsmouth, UK) surrounding the microscope and microfluidics equipment. Images were pro-cessed as previously described[37,39,41].

**Reporting summary**. Further information on research design is available in the Nature Research Reporting Summary linked to this article.

## Data availability
The 3D cryoEM density maps generated in this study have been deposited in the Electron Microscopy Data Bank (EMDB) under accession code EMD-12874. The atomic coordinates have been deposited in the Protein Data Bank (PDB) under accession number 7OFH. The source image data used in this study have been deposited to the Electron Microscopy Public Image Archive (EMPIAR) under accession number EMPIAR-10807. Source data for Fig. 5d are provided with this paper.

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

## Acknowledgements

This research was funded in part by the Wellcome Trust [Grant number 210363/Z/18/Z]. For the purpose of open access, the author has applied a CC BY public copyright licence to any Author Accepted Manuscript version arising from this submission. A Wellcome Trust Seed Award in Science awarded to V.G. (210363/Z/18/Z), along with the University of Exeter, supported R.C. M.M. and K.S. were supported by a BBSRC responsive mode grant awarded to V.G. (BB/R008639/1). U.L. was supported through a BBSRC responsive mode grant (BB/V008021/1), an MRC Proximity to Discovery EXCITEME2 grant (MCPC17189) and an award from the Gordon and Betty Moore Foundation Marine Microbiology Initiative (GBMF5514) awarded to S.P. M.R.L.S. was supported by an Australian Postgraduate Award (APA), PhD scholarship and an IMB Research Advancement Award. B.D. received funding from the European Research Council (ERC) under the European Union's Horizon 2020 research and innovation programme (grant agreement No 803894). We acknowledge Diamond Light Source for access and support of the cryo-EM facilities at the UK's national Electron Bio-imaging Centre (eBIC) at Diamond Light Source [under proposal BI25452], funded by the Wellcome Trust, MRC and BBRSC. We acknowledge access and support of the GW4 Facility for High-Resolution Electron Cryo-Microscopy, funded by the Wellcome Trust (202904/Z/16/Z and 206181/Z/17/Z) and BBSRC (BB/R000484/1). The deposited dataset was collected at eBIC, and the GW4 facility was used for sample screening. We are grateful to Ufuk Borucu of the GW4 Regional Facility for High-Resolution Electron Cryo-Microscopy for help with screening, Kate Heesom of University of Bristol Proteomics Facility for the mass spectrometry

analysis, and to Marjorie Russel for valuable comments on the manuscript, advice, and her generous gifts of the f1pIV expression vectors (pPMR132 and pPMR132$^{S324G}$) and anti-f1pIV antibody.

## Author contributions

R.C. expressed and purified the protein, prepared samples for cryoEM, processed cryoEM data, built the atomic model, interpreted results and prepared figures. M.M. collected and processed cryoEM data. U.L. performed microfluidics experiments and analysed single-cell data. K.S. prepared samples for single-cell microfluidic experiments and supported R.C. S.P. designed single-cell microfluidics experiments. M.R.L.S and M.A.T.B designed and synthesised the macrolide fluorescent probes used in the microfluidics experiments. B.D. interpreted data. J.R. and V.A.M.G. conceptualised the project; V.A.M.G. designed the research, obtained funding for the project and wrote the manuscript with R.C. All authors commented on the manuscript.

## Competing interests

The authors declare no competing interests.
