## [Peer Review File · Nature Communications]

CryoEM structure of the outer membrane secretin channel pIV from the f1 filamentous bacteriophageREVIEWERS' COMMENTS

Reviewer #1 (Remarks to the Author):

This manuscript describes the structure of the secretin protein pIV solved by single particle cryoelectron microscopy at a resolution of 2.7 Å. pIV is a multimeric barrel utilized by filamentous bacteriophages to translocate across the bacterial outer membrane during egress without lysing the cell. Prior studies have described the structure of this secretin at a 22 Å resolution, and a mutational analysis established the importance of the Gate 1 and 2 regions for pore closure. Several other secretins have also been structurally analyzed at this time, which on the one hand diminishes the novelty of this work, but on the other enables detailed comparisons between this phage secretin and the other secretins that function in protein export or pilus biogenesis. Overall, the manuscript is very well developed, including the data presentation and associated text. A fundamental concern is whether the report of another secretin structure, with modeled Gate 1 and 2 domains that were identified previously through mutation and also described structurally for other secretins, advances our understanding of secretin structure and function beyond what was already known. The demonstration that another antibiotic, in this case a hydrophobic macrolide, crosses through the gate-mutated channels, expands on the known list of permeable antibiotics without contributing new insights into the gating mechanism. The proposed model (Fig. 7) is derived from prior structural findings for other secretins, so it is not evident how the present structure better informs this model. Despite these major concerns surrounding lack of novelty and incremental advance to the field, the technical aspects of the study, including data collection and analyses, interpretation and presentation, and overall crafting of the manuscript are extremely well done and solid.

1. Fig. 1. The 15-fold symmetry is not obvious from the images without symmetry imposed, and the image at the far right adds nothing to clarify this symmetry. In counting the number of peripheral densities in the red-boxed image, as well as the image to the left, there are clearly more than 15. This is not to question the 15-fold symmetry of the final 3D reconstruction, just that the fold symmetry is not evident in the panel A images.

2. L. 155. Has it been established that the N0 domain interacts directly with the pI/pXI proteins in the inner membrane, if so, what is the nature of this interaction?

3. L. 212. Fig. 4. The structure of the closed state of the gate region is well-described, as are the structural differences between this gate region and that from InvG secretin. Modeling based on the InvG structure in the open conformation is interesting, but speculative and does not really add much mechanistic insight into how the phage would engage with the secretin or induce structural changes necessary for egress.

4. L. 232. Fig. 5a,b. The mutations conferring leakiness were previously already mapped within residues comprising the gate region or N3 domain, it's not clear what additional information is gained here beyond placing the mutations in a structural context.

5. L. 243. The effects of these mutations in highly conserved Gly and Asn/Arg residues in disrupting the gating would have been predicted on the basis of previous secretin structures, how does the pIV map enhance this understanding?

6. L. 279. The nearly 12 min period for detection of AB uptake is not exactly rapid intracellular accumulation, suggest rephrasing. Perhaps more fundamentally, it is not evident how these findings advance our knowledge of the leaky mutations in conferring gate-opening, especially considering that the MIC is shifted only slightly relative to the WT channel (from 96 to 64 $\mu\text{g/ml}$).

7. L. 348. Fig. 7. This model predicts a mechanism for how the phage might pass through the open pIV channel, but is based on the conserved architecture of the secretin family. Consequently, the model could be envisioned completely independently of the current structure. Yet, in the Abstract, Intro, and Discussion, it is stated that the mechanism for phage-mediated gate movement is based on modeling of the channel opening, which really isn't the case. It's not clear how the structure informs this model.

Reviewer #2 (Remarks to the Author):

This is an excellent paper that was a pleasure to read. The structure will have impact not only in terms of phage biology and bacteriology, but in terms of potential therapeutics. The work appears to have been done carefully, and I had no issues with the structures or the interpretations. I had only the most minor concerns that can be easily addressed:

1) Throughout the paper, such as line 64, the authors talk about monomers in the complex. A monomer is by definition monomeric. They therefore should replace monomers in this context with subunits.

2) The homology model of the N0 domain was created before the AlphaFold server was available. The authors should see what structure is generated using that server.

3) They suggest that the inner pore of the secretin is positively charged, and that this will reduce friction for the negatively charged phage moving through this pore. The opposite would seem to be more likely:

the opposing charges will lead to favorable interactions, which means the phage would stick to the walls. If they had the same charge, there would be repulsion which would reduce the friction from interactions. This has been discussed with respect to DNA passing through the phage T4 tail tube in Zheng et al., *Structure* 25, 1-6 (2017), and DNA passing through mating pili in Costa et al., *Cell* 166, 1436-1444 (2016).

4) In line 382, they say "shows most sequence homology to GspD". This is an improper use of the term homology. Something is homologous (having common ancestry) or not. What they mean is "shows most sequence similarity to GspD".

Point-by-point response to reviewers

We thank the reviewers for their thorough reading of our manuscript, appreciation of the experimentation and writing, and their helpful comments. We have addressed all points in full as explained below. The line numbering in our response refers to the document with track changes.

Reviewer #1 (Remarks to the Author):

*This manuscript describes the structure of the secretin protein pIV solved by single particle cryoelectron microscopy at a resolution of 2.7 Å. pIV is a multimeric barrel utilized by filamentous bacteriophages to translocate across the bacterial outer membrane during egress without lysing the cell. Prior studies have described the structure of this secretin at a 22 Å resolution, and a mutational analysis established the importance of the Gate 1 and 2 regions for pore closure. Several other secretins have also been structurally analyzed at this time, which on the one hand diminishes the novelty of this work, but on the other enables detailed comparisons between this phage secretin and the other secretins that function in protein export or pilus biogenesis. Overall, the manuscript is very well developed, including the data presentation and associated text. A fundamental concern is whether the report of another secretin structure, with **modeled Gate 1 and 2 domains that were identified previously** through mutation and also described structurally for other secretins, advances our understanding of secretin structure and function beyond what was already known. The demonstration that another antibiotic, in this case a hydrophobic macrolide, crosses through the gate-mutated channels, expands on the known list of permeable antibiotics **without contributing new insights into the gating mechanism. The proposed model (Fig. 7) is derived from prior structural findings for other secretins, so it is not evident how the present structure better informs this model.** Despite these major concerns surrounding lack of novelty and incremental advance to the field, the technical aspects of the study, including data collection and analyses, interpretation and presentation, and overall crafting of the manuscript are extremely well done and solid.*

The following 3 bullet points are in response to the general comments which we have highlighted in bold in the above paragraph.

- **“..Gate 1 and 2 domains that were identified previously..”**

We absolutely acknowledge that the positions of the gates were predicted through genetic analysis in Spagnuolo *et al* (which we reference throughout). Whilst the predictions were immensely valuable, they were not entirely accurate. In particular Gate 1 was predicted to start in a region that we can now visualise as being within the inner β -barrel. This demonstrates the importance of combining phenotypic and structural data. What distinguishes the pIV structure from the bacterial secretins is that it can be used to clarify and rationalise a vast amount of f1 and pIV mutant phenotypes – not only those on channel gating, but on phage assembly, egress and also phage display. To demonstrate the advances of our work more clearly, we have modified the text in various places e.g. L483-500 surrounding Fig. 7, and L535-549 surrounding the surface charge of f1pIV.

With reference to the “**modelled..domains**”, we would also like to mention that the ~69 amino acid gate region was built for the vast majority of the protein from our cryoEM map directly. Only 15 residues at the centre of Gate 1 were missing interpretable density. We have made this clearer at L144.

- With reference to experiments on roxithromycin “**..contributing new insights into the gating mechanism**”.

We cannot find an example in the text where we suggest that this was the purpose of this experiment. The main finding is that we have demonstrated that a different class of antibiotic (that is widely used) can pass through the open pIV pore. We have made a small modification to one sentence (L386) to ensure that this is clear.

- “**The proposed model (Fig. 7) is derived from prior structural findings for other secretins..**”

The proposed model is derived in part from our structure, in part from the conserved nature of bacterial secretins, and also from a wealth of experimental data on f1 and pIV, much of which was conducted by a co-author of this paper. We have improved the text in the introduction (L83-87) and in the discussion surrounding Fig. 7 (L483-500) to highlight how a combination of structural and phenotypic findings inform the model. Whilst the model is not based solely on our f1pIV structure, we believe that it is nevertheless useful to include in our manuscript as it has not been presented before. The model will help inform readers who are not wholly familiar with the breadth of the f1 phage and secretin literature.

Specific points:

1. Fig. 1. The 15-fold symmetry is not obvious from the images without symmetry imposed, and the image at the far right adds nothing to clarify this symmetry. In counting the number of peripheral densities in the red-boxed image, as well as the image to the left, there are clearly more than 15. This is not to question the 15-fold symmetry of the final 3D reconstruction, just that the fold symmetry is not evident in the panel A images.

Thank you for pointing this out. The 15-fold symmetry can be interpreted from counting the spokes at the centre of the pore, but we agree that this is not easy to see. We have used different software to open a cross-section of the unsymmetrised cryoEM map, plus labelled the panel differently so that it is easier to see the symmetry based on counting the inner spokes.

2. L. 155. Has it been established that the NO domain interacts directly with the pl/pXI proteins in the inner membrane, if so, what is the nature of this interaction?

Interaction of pIV and pl/pXI was studied by chemical crosslinking and genetic approaches (Feng *et al*, 2002). This demonstrated the presence of a trans-membrane complex comprised of pl and pIV, which requires pXI and protects pl from proteolytic cleavage. Words to this effect have been included in the introduction (L65). Genetic analysis mapped the specific NO domain residues that are key to the interaction with pl/pXI (Russel, M., 1993; Daefler *et al*, 1997), and this has been explained in more detail in the Results (L193-197) and new Supplementary Fig. 5b. This clarification has also improved the Discussion surrounding the model described in Fig. 7 (L483-500), so thank you for pointing out that we had not explained this sufficiently.

3. L. 212. Fig. 4. The structure of the closed state of the gate region is well-described, as are the structural differences between this gate region and that from InvG secretin. Modeling based on the InvG structure in the open conformation is interesting, but speculative and does not really add much mechanistic insight into how the phage would engage with the secretin or induce structural changes necessary for egress.

We appreciate the reviewer's comment that modelling the open state based on InvG is interesting. Modelling will always have an element of speculation; nevertheless we believe that it is still worthy of including in the manuscript. Being as there is a structure of an fd phage available, this allowed us to validate that the dimensions of our open channel would be sufficient to allow phage to egress. This gives us further confidence that the modelling based on InvG is correct. We have made this clearer (L314-316). In addition, modelling the open state allowed us to assess electrostatic interactions and propose a model for egress (Fig. 7). We believe that this does show mechanistic insight as described in the improved text around Fig. 7 (detailed above in response to the general comments, third bullet point). We have also added new discussion surrounding the electrostatic charges (L535-549), which is addressed in more detail in response to point 2 from referee 2.

4. L. 232. Fig. 5a,b. The mutations conferring leakiness were previously already mapped within residues comprising the gate region or N3 domain, it's not clear what additional information is gained here beyond placing the mutations in a structural context.

Mutations conferring leakiness were genetically mapped to a linear protein sequence in the past (Spagnuolo *et al*). Understanding the structure of a protein in three dimensions and at atomic-level detail is a significant advance on mapping to a line of amino acid text. For example, our structure allows many previous genetic studies to be rationalised (e.g. we can now understand how certain mutations cause a leaky effect) and we consider this to be a principle highlight of the work. We have made this clearer at L265-273, and L355-365. This is also elaborated on below (point 5).

5. L. 243. The effects of these mutations in highly conserved Gly and Asn/Arg residues in disrupting the gating would have been predicted on the basis of previous secretin structures, how does the pIV map enhance this understanding?

We discuss the importance of Gly and Asn/Arg residues in two places – with respect to the hydrogen bonding network and gate stability observed from our structure directly (now L259-273), and with respect to the map of leaky mutants (now L355-365).

The Gly and Asn/Arg mutations that we discuss around original L243 (now L265-273) that the reviewer refers to relate to the tight hydrogen-bonding network that we observe (Fig. 4a and b) in the gates. We highlight residues that are conserved in most secretins e.g. Arg 293, conserved in phage secretins only e.g. Asn 269, and unconserved amongst secretins e.g. Asn 295. Being as the hydrogen bonding network differs between all secretins, it is important to highlight the origin of the f1pIV gate stability with the structure. We have re-analysed the hydrogen bonding network, been more specific with our descriptions, and highlighted the role of Asn 269 more clearly. As this residue is conserved for phage secretins only, its role in the hydrogen bonding network could not have been predicted from the available structures.

With respect to the map of leaky mutants at L355-365, the Gly 267 and 297 residues (Gate 1 hinge) are highly conserved between secretins, as we show in Supplementary Fig. 3. We therefore agree that we could predict that they would disrupt gating based on homology. Nevertheless, we believe that it is still valuable to describe the positions of the hinges with respect to the closed and open states of f1pIV, especially considering that this is the first atomic-level structure of a phage secretin. As described above, the residues Asn 269, Arg 293 and Asn 295 form a hydrogen-bonding network that is so far unique to f1pIV. We have made this point clearer in the text to highlight the novelty.

6. L. 279. *The nearly 12 min period for detection of AB uptake is not exactly rapid intracellular accumulation, suggest rephrasing. Perhaps more fundamentally, it is not evident how these findings advance our knowledge of the leaky mutations in conferring gate-opening, especially considering that the MIC is shifted only slightly relative to the WT channel (from 96 to 64 µg/ml).*

Thank you for the suggestion. We have re-phrased the text (L397-398) to state that the accumulation of roxithromycin is significantly higher in cells expressing the mutant than the wild-type, rather than referring to the time taken for uptake. The experiments with roxithromycin were not aimed towards improving our understanding of gate opening, rather to test the uptake of a different class of antibiotic (as described in the response to the general comments above, bullet point 2). With response to the point about MIC, it is important to note that the microfluidics experiments run over the minutes timescale and the MIC experiments on the hours timescale, thus the two outputs cannot be compared directly. The important point to note from the MIC experiments is that cells expressing the f1pIV leaky mutant are killed by a lower dose of antibiotics than cells expressing the wild-type. We have written a sentence to this effect to clarify (L401-402).

7. L. 348. *Fig. 7. This model predicts a mechanism for how the phage might pass through the open pIV channel, but is based on the conserved architecture of the secretin family. Consequently, the model could be envisioned completely independently of the current structure. Yet, in the Abstract, Intro, and Discussion, it is stated that the mechanism for phage-mediated gate movement is based on modeling of the channel opening, which really isn't the case. It's not clear how the structure informs this model.*

Without the f1pIV structure we would not have been able to show the structural rearrangements occurring on channel opening, nor interaction of f1pIV with f1 phage, which forms the framework for the model proposed in Fig. 7. As described in the response to the general comments (bullet point 3), the model is based on our structure, supported by the conserved architecture of the secretin family, and also from phenotypic data specific to f1 and pIV. As already described, we have made the relevance of the model clearer in the text, and have improved Fig. 7. We have also altered wording in the abstract (L23), introduction (L83-87) and discussion (L483-500) to clarify how the model was proposed more clearly.

Reviewer #2 (Remarks to the Author):

This is an excellent paper that was a pleasure to read. The structure will have impact not only in terms of phage biology and bacteriology, but in terms of potential therapeutics. The work appears to have been done carefully, and I had no issues with the structures or the interpretations. I had only the most minor concerns that can be easily addressed:

1) Throughout the paper, such as line 64, the authors talk about monomers in the complex. A monomer is by definition monomeric. They therefore should replace monomers in this context with subunits.

Good point and thank you for mentioning this – we have change instances of monomer.

2) The homology model of the N0 domain was created before the AlphaFold server was available. The authors should see what structure is generated using that server.

We have performed this analysis and now compare all 4 predictions in Supplementary Fig. 5. All structures are in agreement for the folded domain and differ for the linker.

3) They suggest that the inner pore of the secretin is positively charged, and that this will reduce friction for the negatively charged phage moving through this pore. The opposite would seem to be more likely: the opposing charges will lead to favorable interactions, which means the phage would stick to the walls. If they had the same charge, there would be repulsion which would reduce the friction from interactions. This has been discussed with respect to DNA passing through the phage T4 tail tube in Zheng et al., Structure 25, 1-6 (2017), and DNA passing through mating pili in Costa et al., Cell 166, 1436-1444 (2016).

Thank you for pointing this out. We agree that it would make more sense that similar charges would generate electrostatic repulsive forces and thus may reduce friction, as discussed in the above papers. We are also aware of the paper from Yamaguchi *et al*, Nat Commun, (2021), where it was hypothesised that repulsive forces may keep the flagellar rod rotating. However, in their work it was found that both repulsive and attractive forces were important. Being as the f1 phage has a strong negative charge and the centre of f1pIV is more variable, it seems plausible that a mixture of charges is also important for f1 assembly and egress. To address this, we have added discussion regarding the successful use of f1 in phage display, where it has been demonstrated that the f1pIV pore can tolerate a degree variation in the amino acids on the surface of the f1 coat (L535-549).

4) In line 382, they say "shows most sequence homology to GspD". This is an improper use of the term homology. Something is homologous (having common ancestry) or not. What they mean is "shows most sequence similarity to GspD".

Thank you – we have changed the wording.